# Phenotype of Mrps5-Associated Phylogenetic Polymorphisms Is Intimately Linked to Mitoribosomal Misreading

**DOI:** 10.3390/ijms23084384

**Published:** 2022-04-15

**Authors:** Reda Juskeviciene, Ann-Kristina Fritz, Margarita Brilkova, Rashid Akbergenov, Karen Schmitt, Hubert Rehrauer, Endre Laczko, Patricia Isnard-Petit, Kader Thiam, Anne Eckert, Jochen Schacht, David P. Wolfer, Erik C. Böttger, Dimitri Shcherbakov

**Affiliations:** 1Institut für Medizinische Mikrobiologie, Universität Zürich, 8006 Zurich, Switzerland; reda.juskeviciene@gmx.ch (R.J.); mbrilkova@imm.uzh.ch (M.B.); rakbergenov@imm.uzh.ch (R.A.); boettger@imm.uzh.ch (E.C.B.); 2Functional Genomics Center Zurich, ETH Zürich und Universität Zürich, 8057 Zurich, Switzerland; ak.fritz@anatom.uzh.ch (A.-K.F.); hubert.rehrauer@fgcz.ethz.ch (H.R.); endre.laczko@fgcz.uzh.ch (E.L.); david.wolfer@anatomy.uzh.ch (D.P.W.); 3Transfaculty Research Platform Molecular and Cognitive Neurosciences, University Psychiatric Clinics Basel, 4002 Basel, Switzerland; karen.schmitt@upkbs.ch (K.S.); anne.eckert@upk.ch (A.E.); 4Anatomisches Institut, Universität Zürich, 8057 Zurich, Switzerland; isnard@genoway.com (P.I.-P.); thiam@genoway.com (K.T.); 5Kresge Hearing Research Institute, Department of Otolaryngology-Head and Neck Surgery, University of Michigan, Ann Arbor, MI 4605, USA; schacht@umich.edu

**Keywords:** point mutation, mitoribosomal protein, misreading, mitochondria, protein synthesis

## Abstract

We have recently identified point mutation V336Y in mitoribosomal protein Mrps5 (uS5m) as a mitoribosomal *ram* (ribosomal ambiguity) mutation conferring error-prone mitochondrial protein synthesis. In vivo in transgenic knock-in animals, homologous mutation V338Y was associated with a discrete phenotype including impaired mitochondrial function, anxiety-related behavioral alterations, enhanced susceptibility to noise-induced hearing damage, and accelerated metabolic aging in muscle. To challenge the postulated link between Mrps5 V338Y-mediated misreading and the in vivo phenotype, we introduced mutation G315R into the mouse Mrps5 gene as Mrps5 G315R is homologous to the established bacterial *ram* mutation RpsE (uS5) G104R. However, in contrast to bacterial translation, the homologous G → R mutation in mitoribosomal Mrps5 did not affect the accuracy of mitochondrial protein synthesis. Importantly, in the absence of mitochondrial misreading, homozygous mutant *MrpS5*^G315R/G315R^ mice did not show a phenotype distinct from wild-type animals.

## 1. Introduction

Mitochondria are dynamic and multifunctional organelles that produce the majority of cellular ATP and function as a hub modulating cellular homeostasis. Mitochondria are the crossroads of multiple metabolic pathways, including glycolysis, the tricarboxylic acid cycle and fatty acid oxidation; they control calcium signaling and redox state, regulate thermogenesis in warm-blood animals and determine cell fate by integrating death/survival signals. The primary mitochondrial function is to provide eukaryotic cells with energy via oxidative phosphorylation (OXPHOS), which is carried out by four respiratory chain complexes and ATP synthase in the inner mitochondrial membrane [1,2].

Mitochondria are the direct descendants of an alpha-proteobacterial endosymbiont [3]. Over the course of evolution, most of the genome of the bacterial progenitor has been lost or transferred to the nuclear chromosomes, but mitochondria still contain a small circular DNA and protein synthesis machinery required for its expression. Today’s mitochondria contain only a small circular DNA carrying a limited number of genes, i.e., the mammalian mitochondrial genome, approximately 16 kilobases in size, contains 37 genes. Thirteen of them encode protein components of the OXPHOS system and the remaining 24 genes are transcribed into 2 rRNA and 22 tRNA molecules, which maintain mitochondrial translation [4,5]. All the protein constituents of mitochondrial translational apparatus are nuclear-encoded, synthesized on cytosolic ribosomes and imported into the organelle [6].

Little is known about the fidelity of mitochondrial protein synthesis and the effects of mitochondrial mistranslation. Human disease-causing mutations in mitochondrial tRNAs and nuclear genes encoding mitochondrial aminoacyl-tRNA synthetases are associated with severe pathology, known as mitochondrial encephalomyopathies MERRF or MELAS [7,8]. Recently, we reported that mutation V336Y in human mitochondrial ribosomal protein Mrps5 (uS5m) causes error-prone translation of mitochondrial DNA-encoded proteins. Homozygous knock-in mutant mice carrying the mouse homologous Mrps5 V338Y mutation showed impaired mitochondrial function in the brain, decreased neurological stress tolerance, anxiety-related behavioral alterations and accelerated metabolic aging in muscle [9,10,11]. To challenge the postulated mechanistic link between mitochondrial mistranslation and the phenotype observed in *MRPS5*^V338Y/V338Y^ knock-in mice, we decided to introduce a well-characterized bacterial ribosomal ambiguity mutation (*ram*) into the mitochondrial Mrps5 protein as mitochondrial protein synthesis is phylogenetically related to bacterial translation [12]. Unexpectedly, mitochondrial Mrps5 G313R—the human homologue of the bacterial RpsE G104R *ram* mutation [13]—did not confer mistranslation in human mitochondria. Most importantly, homozygous *MRPS5*^G315R/G315R^ mutant knock-in mice showed no noticeable biochemical or behavioral phenotypic alterations. These results strengthen the mechanistic link between mitochondrial mistranslation and phenotype observed in *MRPS5*^V338Y/V338Y^ mice.

## 2. Results

### 2.1. Modelling of Ram Mutations in Bacterial RpsE, Eukaryotic Cytosolic Rps2 and Mitochondrial Mrps5 Ribosomal Proteins

*Ram* mutations, which impair the accuracy of translation, mostly affect the interaction of ribosomal proteins uS4 and uS5. Specifically, weakening of the uS4/uS5 interaction favors the domain closure process (intersubunit rotation) and stabilizes ternary complex binding, thus facilitating incorporation of mismatched near-cognate aa-tRNAs during mRNA decoding [14]. By aligning the protein sequences of bacterial RpsE (uS5), eukaryotic cytosolic Rps2 (uS5c) and mitochondrial Mrps5 (uS5m) [15], we identified mouse cytosolic Rps2 G203R and A226Y and mouse mitochondrial Mrps5 G315R and V338Y as homologous to the established *ram* mutations G104R in *E. coli* [13] and S200Y in yeast [16], respectively (Appendix A).

Modelling the G104R mutation on available 3D crystal structures of *E. coli* protein RpsE (uS5) suggested that substitution of the small neutral glycine located at the RpsD-RpsE (uS4-uS5) interface with a large, positively charged arginine residue would result in steric hindrance disturbing the RpsD-RpsE protein-protein interaction (Appendix A). A similar effect of steric hindrance affecting RpsE C-terminal domain and disturbing RpsD-RpsE protein-protein interaction has been modelled previously for the *E. coli* RpsE A127Y mutation (Appendix A) and the homologous human mitochondrial mutation Mrps5 V336Y [9]. Thus, on the basis of structural modelling, both mutations in RpsE, G104R and A127Y, are predicted to confer misreading, although to different extents.

Structural modelling of the human cytosolic Rps2(uS5c)-Rps9(uS4c) interface suggested that the Rps2-G203R is unlikely to perturb the conformation of Rps2 or to affect the Rps2-Rps9 interface, due to the presence of a cavity on the surface of mammalian Rps9 protein allowing accommodation of the larger arginine residue (Appendix A). This is in contrast to the described Rps2 A226Y mutation [17] which results in steric hindrance between the tyrosine residue and the α-helix in the protein’s C-terminal domain, affecting the conformation of Rps2 and thus its interaction with Rps9 (Appendix A). Based on the modelling results, we hypothesized that in contrast to A226Y, the mutation G203R should not confer a mistranslating phenotype in cytosolic ribosomes of higher eukaryotes.

Mrps5 (uS5m) encoded by the nuclear gene *MRPS5* is the mitochondrial homologue of bacterial RpsE (uS5) and cytosolic Rps2 (uS5c). Modelling the amino acid positions homologous to described *ram* mutations G104R in *E. coli* RpsE and S200Y in yeast cytosolic Rps2, i.e., G313R and V336Y in human Mrps5, revealed that in contrast to *E. coli*, replacement of glycine by arginine at position 313 of the human Mrps5 would not result in steric hindrance at the interface between uS5m and mitochondria-specific protein mS40 (which structurally replaces uS4 in the mitochondrial ribosome) (Appendix A). However, replacement of the small non-polar valine with a large aromatic tyrosine at position 336 would result in steric hindrance within the C-terminal domain of Mrps5 affecting the Mrps5-mS40 interaction (Appendix A). Thus, based on modelling we predicted that in contrast to Mrps5 V336Y, mitochondrial Mrps5 G313R is unlikely to perturb the Mrps5-mS40 interaction and thus should not result in a *ram* phenotype of mitochondrial ribosomes.

### 2.2. Mutations G104R and A127Y in RpsE Confer Misreading in Bacterial Ribosomes

*E. coli* G104R mutant strains were generated by allelic replacement techniques. In addition, we generated merodiploid strains of *Mycobacterium smegmatis* carrying the homologous mutation G129R in RpsE (Appendix A). Previously described bacterial strains carrying A127Y (*E. coli*) or homologous S152Y (*M. smegmatis*) in RpsE were used as positive controls [9]. Ribosomal misreading and read-through were assessed using dual luciferase gain-of-function reporters in cell-free translation assays as described [18]. The dual luciferase translation assays demonstrated that both bacterial RpsE mutations confer significant levels of misreading and read-through (Figure 1a,b). As expected, the mutation-induced effect was limited to misreading of near-cognate codons, but did not extend to non-cognate codons.

### 2.3. Rps2 A226Y, but Not Rps2 G203R, Confers Misreading in Eukaryotic Cytosolic Ribosomes

To challenge the postulated predictions and the possible absence of a *ram* effect for the eukaryotic cytosolic mutation Rps2 G203R, we introduced the mutant Rps2 into HEK293 cells by transient transfection. The established eukaryotic cytosolic *ram* mutation Rps2 A226Y was used as a positive control and Rps2 WT as a negative control. The ability of G203R to induce near-cognate misreading and stop codon read-through was assessed by co-transfection with a vector carrying the dual-luciferase mistranslation reporter system as described previously [17]. As our modelling predicted, we found that the G203R mutant showed no significant differences in misreading and read-through rates when compared with the wild-type (Appendix A). In contrast, the A226Y mutant restored significantly higher Fluc activity in comparison to the wild-type using both the near-cognate misreading and the read-through construct. To exclude possible artifacts related to transient transfection, we generated stably transfected HEK293 cells constitutively expressing mutant G203R or A226Y Rps2 (the latter as a positive control) or wild-type Rps2 (as a negative control). The level of transgenic Rps2 expression in comparison to endogenous Rps2 was monitored by quantitative real time PCR (qRT-PCR) using corresponding Taqman probes. Transgenic mRNA expression levels for G203R and wild-type Rps2 ranged between 35 and 60% of total Rps2 expression and were not statistically different from each other, whereas expression of the Rps2 A226Y was significantly reduced (Appendix A). Consistent with the data from the transient transfections, we found that in contrast to the A226Y clones, G203R mutant clones showed no significant difference in restoring Fluc activity (Appendix A) for both near-cognate misreading and stop codon read-through when compared with the wild-type.

### 2.4. Mitochondrial Protein Synthesis in MRPS5 Mutants

We next generated HEK293 cells stably expressing mutant G313R or wild-type Mrps5. Expression of the transgene was monitored by a GFP reporter located downstream from the gene *MRPS5* and transcriptionally coupled via an internal ribosomal entry site (IRES). GFP fluorescence was similar for wild-type and mutant G313R transfectants (Appendix A). Quantification of *MRPS5* mRNA by qRT-PCR further confirmed that the expression levels of wild-type and G313R mutant transgenes were comparable (Appendix A). Using *myc*-tag fused to the transgenic *MRPS5,* we demonstrated that the protein localizes to the mitochondrial fraction (Appendix A).

To examine the accuracy of mitoribosomal translation we used a previously established *in-organello* translation assay [9]. This assay is based on the ratio of cysteine/methionine (^35^S-Cys/^35^S-Met) incorporation in mtDNA-encoded MT-CO1 protein. As a positive control, we included *MRPS5* V336Y mutant cell lines. In contrast to the V336Y mutation, the G313R mutation in *MRPS5* did not result in increased ratios of cysteine/methionine incorporation in the MT-CO1 protein (Figure 2a). In addition, we analyzed mitochondrial translation accuracy in the *MRPS5* G313R transfected cells using the MT-CO1 *in-organello* translation read-through assay. This assay is based on the observation that mistranslating mitoribosomes can bypass the MT-CO1 stop codon by read-through, thus translating the poly-A sequence and resulting in an extended protein with a C-terminal poly-lysine stretch [9]. Extended MT-CO1 protein labelled with ^14^C-Lys was subsequently collected by immunoprecipitation with anti-poly-lysine antibodies and quantified. In contrast to *MRPS5* V336Y transfected cells, we did not find increased amounts of extended MT-CO1 in mitochondrial extracts of *MRPS5* G313R transfected cells (Figure 2b). These results are consistent with our modelling predictions, which postulate that Mrps5 G313R does not affect the mS40-Mrps5 interface and consequently should not confer a *ram* phenotype.

We next assessed whether the G313R mutation in human Mrps5 protein impacts the general efficiency of mitochondrial protein synthesis or cellular fitness monitored as doubling time. Neither quantitative (^35^S-Met incorporation measured by filter-binding assay) nor qualitative (SDS-PAGE of ^35^S-Met labelled proteins) assessments of mitochondrial translation detected any difference between wild-type and mutant *MRPS5* transfected HEK cells (Figure 2c; Appendix A), nor did the G313R mutation affect the cell’s doubling time (Appendix A).

### 2.5. Transcriptome Profiling of MRPS5 Mutant Cells

Total RNA extracted from HEK293 cells stably transfected with *MRPS5* wild-type or *MRPS5* G313R mutant constructs was subjected to transcriptome profiling by RNA sequencing. Previously published mutant *MRPS5* V336Y transfected cells [9] were included in the analysis for comparison. RNAseq revealed 921 differently expressed genes (DEGs) between *MRPS5* G313R and WT cells (*p* < 0.05, no fold-change threshold) and 2659 DEGs between *MRPS5* V336Y and WT cells (*p* < 0.05, no fold-change threshold).

Identified DEGs were subjected to Gene Ontology (GO) term enrichment analysis using the free online tool EnrichR, [19] and process network analysis using the commercial tool MetaCore, “MetaCore. Available online: https://clarivate.com/cortellis/solutions/early-research-intelligence-solutions/ (accessed on 1 April 2022)”. No significantly regulated pathways were identified in the G313R dataset compared with the V336Y dataset in which translation-related terms were found to be significantly enriched among the elevated transcripts (Figure 2d,e).

Interestingly, enrichment analysis using GO Cellular Component terms suggested that in addition to the upregulation of the cytosolic ribosome components “large ribosomal subunit” (*p*_adj_ = 9.7 × 10^−3^) and “ribosome” (*p*_adj_ = 1.1 × 10^−2^), mitochondrial function is affected in V336Y mutant cells. Four mitochondria-associated GO categories were identified among upregulated DEGs: “mitochondrial ribosome” (*p*_adj_ = 1.95 × 10^−2^), “mitochondrial matrix” (*p*_adj_ = 1.95 × 10^−2^), “mitochondrial respiratory chain complex I” (*p*_adj_ = 3.8 × 10^−2^), and “mitochondrial inner membrane” (*p*_adj_ = 4.7 × 10^−2^). In total, 70 transcripts of mitochondrial proteins were significantly increased in V336Y mutant cells. Analysis of functional annotations revealed that the two largest groups of transcripts were related to mitochondrial translation (15 genes) and mitochondrial respiration (13 genes) (Appendix A). In contrast, only 22 transcripts of mitochondrial proteins were found to be significantly (*p* < 0.05) increased in G313R mutant cells compared with WT control cells; among them, five were translation-related and five were components of mito-respiration. No mitochondria-associated GO categories were identified at the level of significance *p*_adj_ < 0.05 in G313R mutant cells.

### 2.6. Generation of MRPS5^G315R/G315R^ Knock-In Mice

To study the in vivo phenotype of the Mrps5 G313R mutation, we constructed a homozygous knock-in mutant *MRPS5*^G315R/G315R^ mouse line (mouse G315R is homologous to human G313R, Appendix A) expressing mutant G315R Mrps5 mRNA (for further details, see Methods and Appendix A). Chimeras were generated using embryonic stem cells derived from Sv129 mice and were then consecutively backcrossed to C57BL/6 mice. After four generations of backcrossing, approximately 96% of the genetic background of the resulting mice was of C57BL/6 origin. In addition, as per selection for the mutant *MRPS5* gene locus during breeding, the genetic material of Sv129 is located in particular around the *MRPS5* gene. Thus, standard procedures of transgenic mice generation bear the inherent risk that the genomic region flanking the mutation of interest, which is carried from Sv129 background during backcrossing, might affect the phenotype [20]. Analysis of RNA sequencing data from brain samples of *MRPS5*^G315R/G315R^ mice (three months old) revealed that an approximately 50 Mb fragment around *MRPS5* contained genes with SNPs characteristic for the Sv129 strain (Figure 3a). *MRPS5*^V338Y/V338Y^ mice, which were generated following the same procedure [9], had an approximately 25 Mb fragment of Sv129 genetic background spanning the *MPRS5* locus in their genome (Figure 3b).

### 2.7. ETC Function and Noise-Induced Auditory Damage in MRPS5^G315R/G315R^ Mutant Mice

Mitochondrial mistranslation in *MRPS5*^V338Y/V338Y^ mice is associated with a decline in OXPHOS function and an increased susceptibility to noise-induced auditory damage [9]. Therefore, we tested whether the function of the mitochondrial respiratory chain and susceptibility to noise-induced auditory damage are affected in *MRPS5*^G315R/G315R^ mice.

The capacity of the OXPHOS system in brain cortical mitochondria of mature (9 months old) and aged (19 months old) *MRPS5*^G315R/G315R^ mutant and age-matched control wild-type mice was evaluated using the Seahorse XF24 flux analyzer system (Figure 4a–d). The oxygen consumption rate (OCR) was similar in the mitochondria of both 9 and 19 months old *MRPS5*^G315R/G315R^ mice compared with age-matched wild-type control mice. More specifically, basal respiration (state 2), maximal respiration (state 3) and uncoupled respiration (state 3u) rates showed the expected age-related decline both in WT and mutant mice, but no significant differences between wild-type and age-matched G315R mutant animals were observed (Figure 4a–d). Additionally, we measured the activities of complex II and complex IV in mitochondria of 19-month-old animals. Complex IV, which contains several mtDNA-encoded subunits, is sensitive to defects in mitochondrial translation, whereas complex II contains only nDNA-encoded subunits and is thus not directly affected by mitochondrial mistranslation. Complex IV/II ratios showed no significant difference between wild-type and mutant animals, indicating that mutation Mrps5 G315R does not disturb mitochondrial translation (Appendix A). Additional markers of mitochondrial functionality, i.e., reactive oxygen species (ROS) production and ATP levels, likewise did not reveal any significant difference between G315R mutant and WT mice (Appendix A).

Next, we examined the auditory function of *MRPS5*^G315R/G315R^ mice using auditory brain stem response measurements. Baseline thresholds of around 20 dB sound pressure level (SPL) were similar for mutant and wild-type animals. For the study of noise-induced auditory damage, ABR thresholds were registered three weeks after the exposure to noise (1–10 kHz broadband at intensities of 108 and 110 dB) when the damage to the auditory system was stabilized. The noise exposure led to increased ABR thresholds at 24 and 32 kHz. However, no significant difference between *MRPS5* mutant and control mice was observed (Figure 4e) indicating that the mutation did not affect auditory function.

### 2.8. Absence of Behavioral Changes in MRPS5^G315R/G315R^ Mice

A set of discrete phenotypic traits have been found to be altered in *MRPS5*^V338Y/V338Y^ mice. In particular, *MRPS5* V338Y mice exhibited increased anxiety-related behavioral changes and impaired learning under stressful conditions [9]. To assess whether the Mrps5 G315R mutation induces behavioral alterations, we conducted a battery of behavioral tests on 9 and 19-month-old animals; some tests were also recorded at 14 months of age. In contrast to *MRPS5*^V338Y/V338Y^ mice, the *MRPS5*^G315R/G315R^ mutants did not exhibit increased anxiety or changes in exploratory behavior compared to age-matched wild-type control mice, as indicated by avoidance of the center in the large open field, time spent in the open arms of the elevated O-maze, and exploratory activity in IntelliCage (Figure 5a,e).

Learning and memory were assessed in place (Figure 6a,b) and cue (Figure 6c) navigation tasks in the water-maze, spontaneous alternation tests in the T-maze (Figure 6d), and in the serial reversal and chaining tasks in IntelliCage (Figure 6e,f). Performance of *MRPS5*^G315R/G315R^ mutant mice was indistinguishable from wild-type controls in all parameters at both 9 months and 19 months of age. Mature and aged mutant mice also showed normal motor function in the following tests: locomotor tests (Appendix A); swim speed (Appendix A); rotarod (Appendix A); and muscle-strength grip test (Appendix A). Hippocampus-dependent burrowing and nesting activities were also not affected by the mutation (Appendix A). Taken together, the behavioral assessments did not show any significant abnormality in the *MRPS5*^G315R/G315R^ animals.

### 2.9. Absence of Metabolic Changes in Skeletal Muscle of MRPS5^G315R/G315R^ Mice

We performed metabolic profiling of skeletal muscle tissue from 9 and 19-month-old *MRPS5*^G315R/G315R^ mice to compare it with the profile of previously studied *MRPS5*^V338Y/V338Y^ mice [10]. In contrast to the V338Y mutation, the G315R mutation did not accelerate age-associated metabolic changes in the muscle tissue (Figure 7a,b). Whereas 9 and 19-month-old animals of all three genotypes (WT, G315R, V338Y) were clearly separated along axis 1 of the BGA scatter plot, which correlates with aging, 19-month-old G315R mutants had a metabolite profile close to that of 19-month-old wild-type controls. In contrast, the metabolic profile of 19-month-old V338Y mutants was significantly (*p* < 0.0001) shifted further along axis 1, indicating increased metabolic aging in those mutants. The conclusion that *MRPS5*^G315R/G315R^ animals, unlike *MRPS5*^V338Y/V338Y^ mice, do not show accelerated metabolic aging in muscle tissue, was further corroborated by calculating the BGA on the basis of a previously described dataset [21] and projecting our animals on this model using a dataset of 88 common metabolites (Figure 7b).

## 3. Discussion

The impact of mitochondrial dysfunction on organismal health has been studied since the early 1960s when mitochondrial myopathies were first described [22]. Mostly, attention had focused on uncommon, severe OXPHOS defects which often result in highly debilitating or even fatal disease [23]. However, accumulating evidence suggests that mitochondria are also involved in the pathogenesis of multiple common disorders, including diabetes [24], obesity [25,26], and neurodegeneration [27].

Mitochondrial mistranslation may affect mitochondrial function and thus contribute to the development of mitochondria-related pathologies. Recently, we have reported the first mitochondrial ribosomal ambiguity (ram) mutation V338Y in mouse mitochondrial ribosomal protein S5 (Mrps5), which is homologous to human Mrps5 V336Y. We found that V338Y-mediated error-prone mitochondrial translation reduces OXPHOS capacity and is associated with behavioral alterations [9] and accelerated age-related metabolic changes in muscle [10] of mutant mice.

Here, we challenged the mechanistic link between in silico modelling, mitochondrial mistranslation and mitochondria-related pathology observed in the knock-in transgenic *MRPS5*^V338Y/V338Y^ mice. For this, we focused on *MRPS5* mutation G315R (G313R in human mitochondrial ribosomes), which is homologous to *RpsE* G104R, a well-established bacterial *ram* mutation. However, structural modelling suggested that the glycine to arginine substitution in this position leads to steric hindrance in bacterial but not in mitochondrial ribosomes. In agreement with the *in silico* based predictions, bacterial RpsE G104R mutant ribosomes demonstrated elevated level of mistranslation, whereas mitochondrial Mrps5 G313R ribosomes did not show impaired accuracy of mitochondrial translation *in organello*. In other words, although the mutation Mrps5 G313R has been identified as homologous to the described bacterial RpsE G104R *ram* mutation, in contrast to RpsE G104R, the mutation Mrps5 G313R does not confer misreading. Here we have demonstrated both effects with a bacterial in vitro system and a mitochondrial *in organello* system and shown that the discrepancy can be explained on the basis of structural modelling. To support this approach, we have shown that the mistranslation caused by the described *ram* mutations RpsE A127Y (*E. coli*) and Mrps5 V336Y (human mitochondria) is in agreement with the results of structural modelling.

Subsequently, we found that transcriptome profiling of *MRPS5*^G313R^ mutant cells does not reveal any significantly enriched term compared to the *MRPS5*^V336Y^ mutation which is associated with an enrichment of transcripts related to translation and mitochondrial function. Increased expression of cytosolic ribosomal proteins has previously been suggested as a compensatory mechanism for the mitochondrial dysfunction caused by the mitochondrial 12S rRNA A1555G mutation [28]. Thus, the transcriptome profiling further corroborates that V336Y, but not G313R mutant cells, show evidence of mitochondrial dysfunction associated with error-prone mitochondrial translation. The elevated expression of mitochondria-related genes observed in the V336Y mutants, including genes coding for OXPHOS subunits and components of mitoribosomal translation machinery, most likely reflects a known compensatory response to mitochondrial dysfunction [29].

Our data obtained with transfected HEK293 human cell lines provide strong evidence that Mrps5 G313R, in contrast to V336Y, is not a mitochondrial *ram* mutation. To further challenge the link between V336Y-mediated mitochondrial misreading and the phenotype observed in transgenic *MRPS5*^V338Y/V338Y^ mice, we generated homozygous knock-in mice *MRPS5*^G315R/G315R^. In particular, we wished to exclude (i) that mutations in *MRPS5* affect mitochondrial function independently of misreading, and (ii) that it is not the V338Y mutation, but the surrounding Sv129 genomic locus which is responsible for the *MRPS5*^V338Y/V338Y^ mice phenotype. As per classical transgenic mice technology, the *MRPS5* mutations were introduced into embryonic stem (ES) cells of Sv129 genetic background and then transferred to a C57BL/6 background by successive backcrossing. Inherently, the genomic region flanking the targeted gene in mutant mice (but not in the WT littermate control) was carried from the Sv129 background during backcrossing and might affect the phenotypic outcome [30]. RNA-seq data revealed that the size of the Sv129-derived donor region in *MRPS5*^V338Y/V338Y^ and *MRPS5*^G315R/G315R^ mice was similar, with approximately 25 Mb and 50 Mb, respectively.

*MRPS5*^G315R/G315R^ mice were subjected to a set of behavioral tests similar to that used previously for characterization of the *MRPS5*^V338Y/V338Y^ mice [9]. Our investigations in G315R mice did not show any of the phenotypic changes characteristic of the *MRPS5*^V338Y/V338Y^ mutant mice, such as stress intolerance or anxiety-related behavioral alterations, nor any other phenotypic trait different from the WT controls. In addition, neither did we observe increased susceptibility to noise-induced hearing loss. Importantly, we also did not find any effect of the Mrps5 G315R mutation on mitochondrial respiration, ATP levels or generation of ROS. Furthermore, metabolic profiles of skeletal muscle tissue from *MRPS5*^G315R/G315R^ mice were not significantly different from those of wild-type mice. Besides testifying to the specific effects of the V338Y mutation, the absence of phenotypic alterations in the G315R mutants effectively rules out the possibility that the Sv129 genomic locus encompassing the Mrps5 mutation confers the phenotypic alterations observed in the V338Y mutants.

In conclusion, we here provide evidence that mutation Mrps5 G313R, the mitochondrial homologue of the bacterial RspE G104R *ram* mutation, does not affect the accuracy of mitochondrial translation and does not show any of the phenotypic changes observed in *MRPS5*^V338Y/V338Y^ transgenic mice. Our results strengthen the pathomechanistic link between error-prone mitochondrial translation, impaired mitochondrial respiration, enhanced susceptibility to noise-induced hearing damage, anxiety-related behavioral alterations, and accelerated age-related metabolic changes in the muscle of *MRPS5*^V338Y/V338Y^ mice.

## 4. Materials and Methods

### 4.1. Bacterial Strains

Generation of merodiploid *M. smegmatis* strains *rpsE* G129R and *rpsE* WT was performed as previously described for the strain *rpsE* A152Y [9]. In brief, genomic DNA from *M. smegmatis* served as a template for *rpsE* gene amplification. Fusion PCR via complementary overhanging sequences was used for gene construction. The constructs were cloned into integrative vector pMIH using restriction enzymes HindIII and SpeI (Thermo) resulting in vectors pMIH-rpsE-G129R or pMIH-rpsE-WT. The vectors were electroporated into *M. smegmatis* mc^2^155 *ΔrrnB* [31]. Single colonies were picked, propagated for further analysis and checked by PCR and DNA sequence analysis.

Generation of *E. coli* strains *rpsE* G104R and *rpsE* WT using oligo λ Red-mediated recombination was performed as described previously [32]. In brief, two steps of recombineering were conducted, using *E. coli* MG1655 and oligonucleotides that comprised the G104R mutation or fragment of *rpsE* WT. Positive clones were screened by PCR and confirmed by DNA sequencing. The recombinant *E. coli* strains were a kind gift of Diarmaid Hughes and Douglas Huseby, Uppsala University, Sweden.

For the growth experiments, bacterial strains were streaked on agar plates and grown at 37 °C until single colonies were visible. Single colonies were used to inoculate liquid LB medium. Growth experiments were started at an initial OD_600_ of 0.002 and further incubated at 37 °C. OD_600_ for *E. coli* was measured every 30 min. Signal intensities at time point 0 h were set as one and growth curves were plotted. Doubling time was calculated as t_D_ = ln(2)/slope.

### 4.2. Cell Culture and Transfection of HEK293 Cells

Human embryonic kidney cells (HEK293, Innoprot) were maintained in complete Dulbecco’s Modified Eagles Medium (DMEM) (Life Technologies) containing 10% Fetal Bovine Serum (FBS) (Life Technologies), at 37 °C in 5% CO_2_.

Plasmid pMouse*MRPS5*-WT which contains the mouse *MRPS5* coding region under the control of the chicken β-actin promotor and the CMV enhancer (CAGGS promotor) was constructed by Genoway (Lyon, France). In brief, the mouse *MRPS5* wild-type CDS was synthesized, and the restriction sites BamHI and AvrII were introduced at the 5′ and the 3′ end. Using the BamHI and AvrII restriction sites, the 1318 nt sized fragment was ligated into a eukaryotic expression vector under control of the CAGGS promotor and the human growth hormone polyadenylation signal, resulting in pMouse*MRPS5*-WT.

The plasmid expressing the human *MRPS5* gene was constructed as follows. A cDNA fragment containing human *MRPS5* wild-type CDS was generated by PCR, using cDNA reverse transcribed from total RNA extracted from HEK293 cells. Total RNA was prepared using Trizol (Life Technologies) and DNAse treated (DNaseI, Life Technologies) by incubation for 15 min at 37 °C; 1 µL of 25 mM EDTA was added and samples were incubated for 10 min at 65 °C and stored at −80 °C. RNA was reverse transcribed using a High Capacity RNA to cDNA kit (Life Technologies) according to the manufacturer’s instructions. V336Y (GTC → TAC) and G313R (GGG → GGC) point mutations were introduced by PCR-mediated site-directed mutagenesis, and restriction sites for BamHI and XhoI were introduced at the 5′- and 3′-end of the fragments, respectively. The mouse *MRPS5* wild-type CDS in pMouse*MRPS5*-WT vector was replaced with human *MRPS5* wild-type, human *MRPS5* V336Y or human *MRPS5* G313R using BamHI/XhoI restriction sites, resulting in vectors ph*MRPS5*WT, ph*MRPS5*V336Y and ph*MRPS5*G313R. IRES-eGFP was amplified from plasmid pLZRSpBMN-IRES-eGFP (a kind courtesy of Saule Zhanybekova, University of Basel, Switzerland), and inserted downstream of the *MRPS5* CDS using restriction sites for BstBI and ClaI, resulting in vectors ph*MRPS5*WT-IRES-eGFP, ph*MRPS5*V336Y-IRES-eGFP and ph*MRPS5*G313R-IRES-eGFP. The hygromycin B resistance cassette was amplified by PCR from vector pGL4.14 (Promega), subcloned into pGEM^®^-T Easy and inserted into ph*MRPS5*WT-IRES-eGFP, ph*MRPS5*V336Y-IRES-eGFP and ph*MRPS5*G313R-IRES-eGFP using MluI specific restriction sites.

For generation of stable transfectants, HEK293 cells were transfected with vectors ph*MRPS5* WT-IRES-eGFP or ph*MRPS5*-G313R-IRES-eGFP using Turbofect (Thermo Fisher, Waltham, MA, USA) according to the manufacturer’s instructions. Cells were cultured in DMEM supplemented with 10% FBS and propagated under hygromycin B (100 µg/mL) selection for 5–7 weeks. GFP expressing colonies were selected for further characterization. GFP fluorescence was analyzed by flow cytometry using the BD FACS Canto II (BD Biosciences) and the FlowJo data analysis software (BD Biosciences).

### 4.3. Determination of MRPS5 mRNA Expression

Quantitative real-time PCR was used to determine *MRPS5* transgene mRNA expression relative to endogenous *MRPS5* mRNA levels as previously described [9].

The ratio of *MRPS5* transgene mRNA to endogenous *MRPS5* mRNA for cell lines expressing mutant G313R *MRPS5* was determined using *MRPS5* forward 5′-CTG CCA CAG GGC CAT CAT CAC CAT CTG C-3′ and *MRPS5* reverse 5′-CGG AAG AGG CCC TGG GTG AGG CTG AGC-3′ primers flanking the site of mutation. These primers amplify a 108 nt PCR fragment of both endogenous and transgene *MRPS5* mRNA. Discrimination was achieved by using TaqMan probes which recognize specifically human wild-type endogenous *MRPS5* (5′- CAA AGG TTA CGG CCT -3′, conjugated to NED) or human mutant transgene *MRPS5* G313R (5′- AAA GGT TAC AGG CTC -3′, conjugated to FAM). For each sample, TaqMan real-time qPCR was conducted in triplicates using the TaqMan Kit (Life Technologies) and the ABI 7500 Fast Real-Time PCR System (Life Technologies). Amplification consisted of 40 cycles (95 °C for 20 s and 60 °C for 45 s), the ratio of transgene versus endogenous *MRPS5* was calculated as described previously [33].

### 4.4. Cellular Localization of the Transgenic MrpS5 Protein

To monitor cellular localization of the Mrps5 protein, we introduced a 33 nt fragment GAG CAA AAG CTC ATT TCT GAA GAG GAC TTG AAT coding for *myc*-tag peptide EQKLISEEDLN at the C-termini of both *MRPS5* WT and *MRPS5* G313R by PCR, resulting in vectors ph*MRPS5*WTmyc-IRES-eGFP and ph*MRPS5*G313Rmyc-IRES-eGFP. HEK293 cells were transiently transfected using TurboFect and cultured in DMEM supplemented with 10% FBS for 72 h. Cellular localization of Mrps5 protein was assessed by a Western blot of mitochondrial and cytosolic fractions using anti-*myc* antibody (ab9106; Abcam).

### 4.5. Determination of Generation Time in HEK293 Cells

Cellular growth was monitored using Alamar Blue (Life Technologies, Carlsbad, CA, USA). HEK293 cells were seeded on 24 well plates (BD Falcon) at low density and incubated in DMEM with 10% FBS at 37 °C. 10% Alamar Blue was added to the cells (*v*/*v*) at time points 0, 24, 48 and 72 h, and fluorescence was measured (excitation: 530 nm, emission: 590 nm) after 3 h incubation. Signal intensities at time point 0 h were set as one and growth curves were plotted. Doubling time was calculated as t_D_ = ln(2)/slope.

### 4.6. Cell-Free Luciferase Translation Assays, Mitochondrial Translation Assays, Immunoprecipitation Analysis and Western Blots

These analyses were performed as previously described [9].

### 4.7. Generation of Knock-In MRPS5 Transgenic Mice

*MRPS5* G315R mice were generated using a FLEx approach as previously described [9]. In brief, the gene-targeting vectors were constructed from genomic 129Sv mouse strain DNA. Linearized targeting vectors were transfected into 129Sv ES cells, positive selection was started 48 h after electroporation by the addition of G418. The resistant clones were isolated, amplified, duplicated and genotyped by both PCR and Southern blot analysis. PCR and Southern blot genotyping led to the identification of two targeted clones for *MRPS5* G315R.

Recombined ES cell clones were microinjected into C57BL/6 blastocysts and gave rise to male chimeras. Breeding with C57BL/6 Flp deleter mice (CAG-Flp) was performed to produce the *MRPS5* heterozygous inducible line devoid of neomycin cassette: *MRPS5*^FLEx/WT^. Heterozygous inducible mice (*MRPS5*^FLEx/WT^*)* were first identified by PCR. PCR positive mice (*MRPS5*^FLEx/WT^*)* were confirmed by Southern blot analysis. Inducible mice were backcrossed to C57BL/6 for four generations. The inducible *MRPS5* G315R line was crossed to C57BL/6 CRE deleter mice (CMV-Cre) to produce the *MRPS5* G315R heterozygous induced-mutant line. Cre-recombination resulted in flipping of the duplicated inverted exon bearing the mutation of interest (FLEx strategy) and expression of the mutant allele, resulting in the heterozygous mutant line *MRPS5*^G315R/WT^ referred to as induced mice. For each line, heterozygous mice were genotyped by PCR, Southern blot and sequencing (Appendix A).

Heterozygous *MRPS5*^G315R/WT^ mice were viable, fertile and did not show any gross phenotype. Heterozygous mutant mice bred within the expected Mendelian ratio, suggesting that the mutation has no significant effect on the viability of the line. Interbreeding *MRPS5*^G315R/WT^ resulted in homozygous *MRPS5*^G315R/G315R^ mice. Homozygous mutant mice were viable, able to breed and did not show any gross phenotype. Analysis of *MRPS5* gene expression in the homozygous mutant mice revealed *MRPS5* mutant mRNA only in all tissues tested (Appendix A). The overall expression level of mutant *MRPS5* mRNA was comparable to that of wild-type mice, indicating that the mutation did not lead to deregulation of *MRPS5* gene expression.

For the RNA extraction, mice tissue sections were collected and quick frozen. RNA was extracted using Trizol (Life Technologies) according to the manufacturer’s protocol. cDNA synthesis was performed using a SuperScript II cDNA synthesis kit (Life technologies) using random hexamers and 40 ng RNA in 20 µL reagent according to the manufacturer’s protocol. Expression of the mutant gene was assessed from the brain, inner ear, and retina recovered from 6 to 7-week-old animals. Heterozygous *MRPS5* induced mice (*MRPS5*^G315R/WT^) were compared to wild-type mice for expression of the *MRPS5* gene. RT-qPCR analysis showed no significant difference in *MRPS5* total mRNA between heterozygous induced mutant mice and controls. Expression of the G315R mutant allele was demonstrated by sequencing of the RT-PCR product. Primers used for RT-PCR and sequencing were EBO4-SeqF1 (5′-GCT ATT GGG AAA GCT GCT GA-3′) and EBO4-SeqR2 (5′-TCA CGT CCT GCC AGT CCA GC-3′). Primers used for RT-qPCR analysis were EBO4-TOT-F1 (5′-CCA TGA ACA TGC TCA ACC TC-3′) and EBO4-TOT-R2 (5′-ATA GGC AGA GGC CCA CAT T-3′).

### 4.8. RNA Sequencing and Data Analysis

RNA sequencing (RNAseq) was performed at the UZH/ETH Functional Genomics Center Zurich “FGCZ. Available online: http://www.fgcz.ch/. (accessed on 1 April 2022)” according to the Illumina RNA sequencing protocol. RNA was extracted from cultured cells using Trizol. The quality of the isolated RNA was determined using a Qubit^®^ (1.0) Fluorometer (Life Technologies) and a Bioanalyzer 2100 (Agilent). Only those samples with a 260 nm/280 nm ratio between 1.8 and 2.1 and a 28S/18S ratio within 1.5–2 were further processed. The TruSeq Stranded mRNA Sample Prep Kit (Illumina) was used in the succeeding steps. Briefly, total RNA samples (100–1000 ng) were ribosome depleted and then reverse-transcribed into double-stranded cDNA with actinomycin added during first-strand synthesis. The cDNA samples were fragmented, end-repaired and polyadenylated. TruSeq adapters containing the index for multiplexing were ligated to the fragmented DNA samples. Fragments containing TruSeq adapters on both ends were selectively enriched with PCR. The quality and quantity of the enriched libraries were validated using Qubit^®^ (1.0) Fluorometer and the Caliper GX LabChip^®^ GX (Caliper Life Sciences, Waltham, MA, USA). The product is a smear with an average fragment size of approximately 360 bp. The libraries were normalized to 10 nM in Tris-Cl 10 mM, pH 8.5 with 0.1% Tween 20. The TruSeq SR Cluster Kit v4-cBot-HS (Illumina) was used for cluster generation using 8 pM of pooled normalized libraries on the cBOT. Sequencing was performed on the Illumina HiSeq 2500 single end 126 bp using the TruSeq SBS Kit v4-HS (Illumina).

The quality of the reads was assessed using FastQC (Babraham Bioinformatics) “FastQS. Available online: http://www.bioinformatics.babraham.ac.uk/projects/fastqc/. (accessed on 1 April 2022)”, and potential contaminations were evaluated with FastQ Screen (Babraham Bioinformatics) “FastQS. Available online: http://www.bioinformatics.babraham.ac.uk/projects/fastq_screen/. (accessed on 1 April 2022)” using bowtie2 v. 2.1.0 [34] default parameters. Quantification of gene expression was performed using the RSEM package (version 1.2.18) “RSEM. Available online: http://bmcbioinformatics.biomedcentral.com/articles/10.1186/1471-2105-12-323. (accessed on 1 April 2022)” [35], mapping against the ensembl 75 annotations derived from the human genome assembly GRCh37 and mouse genome assembly GRCm37. Genes not present (<10 counts per gene) in at least 50% of samples from one condition were discarded from further analyses. Single nucleotide polymorphisms (SNPs) were called using the Genome Analysis Toolkit (GATK) “GATK. Available online: https://software.broadinstitute.org/gatk/. (accessed on 1 April 2022)”. SNP annotation was conducted with SnpEff [36]. Differential gene expression analysis between sample groups of interest was performed using the R/bioconductor package edgeR [37]. To evaluate functional activities, differentially expressed genes were mapped to known biological ontologies based on the GO project “Gene Ontology. Available online: http://www.geneontology.org/. (accessed on 1 April 2022)” using the gene annotation tool Enrichr [19]. For transcriptome analysis of HEK293 cells, six clones of *MRPS5* G313R or six clones of *MRPS5* V336Y and six clones of *MRPS5* WT stably transfected cells were compared.

Transcriptome data are available in the Gene Expression Omnibus (GEO), accession number GSE195772, token cfwfiksadvgpbgb.

### 4.9. Metabolome Analysis

For metabolome analysis of quadriceps muscle, the following female mice were compared: five 9-month-old animals of *MRPS5*^WT/WT^, three 9-month-old animals of *MRPS5*^V338Y/V338Y^, three 9-month-old animals of *MRPS5*^G315R/G315R^, seven 19-month-old animals of *MRPS5*^WT/WT^, four 19-month-old animals of *MRPS5*^V338Y/V338Y^, and five 19-month-old animals of *MRPS5*^G315R/G315R^ (Appendix A). Metabolome analysis was performed by Metabolon (USA) according to published methods [38]. Samples were prepared by a proprietary series of organic and aqueous extractions in order to remove proteins and to recover the maximum amount of small molecules. The extracted samples were split into equal parts and analyzed via GC-MS or LC-MS/MS. For the LC-MS/MS, two equal parts were analyzed in the positive (acidic solvent) and in the negative (basic solvent) ionization mode. Samples for GC-MS were bistrimethyl-silyl-trifluoroacetamide derivatized and were run with a 5% diphenyl/95% dimethyl polysiloxane fused silica column.

Metabolome data sets were analyzed by Between Group Analysis (BGA), which is based on principal component analysis (PCA) [39], using the made4 package (bioconductor.org) in R. Normalized MS intensity data were used. BGA was applied to a data set consisting of 230 metabolites. Results of the BGA were visualized by a scatter plot for the first two axes of the BGA. Coordinates were used to calculate significance levels between groups by applying Welch’s *t*-test. A common dataset of 88 metabolites derived from the comparison of a published dataset [21] and our dataset was used to compare age associated metabolic patterns. The animals of the published dataset were used as training dataset (3-month-old animals *n* = 8, 23-month-old animals *n* = 7), and a BGA model was calculated. The animals of the present study were used as test dataset and projected on the calculated BGA model. The resulting coordinates of individual mice were plotted. For univariate analyses, Welch’s *t*-test and two-way ANOVA were used.

### 4.10. Cortical Brain Homogenate Preparation

Isolated cortical hemispheres were dissected on ice, washed in ice-cold buffer 1 (138 mM NaCl, 5.4 mM KCl, 0.17 mM Na_2_HPO_4_, 0.22 mM K_2_PO_4_, 5.5 mM glucose, 58.4 mM sucrose, pH 7.35), and homogenized with a glass homogenizer (10–15 strokes, 400 rpm) in 2 mL of ice-cold buffer 2 (210 mM mannitol, 70 mM sucrose, 10 mM Hepes, 1 mM EDTA, 0.45% BSA, 0.5 mM DTT, 5× Complete Protease Inhibitor (*Roche Diagnostics*)).

### 4.11. Preparation of Isolated Mitochondria

Cortical brain homogenates were centrifuged at 1450× *g* for 7 min at 4 °C to remove nuclei and tissue particles, centrifugation was repeated with the supernatant fraction for 3 min [40]. The resulting supernatant fraction was centrifuged at 10,000× *g* for 5 min at 4 °C to pellet mitochondria. The resulting pellet was resuspended in 1 mL of ice-cold buffer 2 and centrifuged at 1450× *g* for 3 min at 4 °C to remove debris. The mitochondria-enriched supernatant was centrifuged at 10,000× *g* for 5 min at 4 °C to obtain the mitochondrial fraction. This fraction was resuspended in 300 μL of PBS and stored at 4 °C until use, followed by determination of protein content.

### 4.12. Oxygen Consumption Measurements

In isolated brain mitochondria, complex IV and complex II measurements in isolated brain mitochondria, and determination of superoxide anion radicals were performed as previously described [9].

### 4.13. Noise Exposure and Auditory Brain Stem Response

Male C57/BL6 mice were first exposed for two hours to broadband noise (2–10 kHz) at intensities of 104, 106, 108 and 110 dB. ABR thresholds were determined three weeks later when damage to the auditory system had stabilized and was no longer confounded by transient impairment of sound processing. Exposures at noise intensities of 108 and 110 dB resulted in robust elevations of ABR thresholds and were chosen as treatment for mutant *MRPS5*^G315R/G315R^ mice and littermate controls. Sham-exposed controls (two hours in the exposure chamber without sound) showed no effect on ABR.

### 4.14. Animals for Behavioral Analysis

Mice were housed under a 12/12h light-dark cycle (lights on at 20:00) in groups of 2–5, unless individual housing was required by experimental protocols. Mice were tested during the dark phase of the cycle under indirect dim light (ca. 12 lux) unless specified otherwise. Before behavioral testing, each mouse was tagged by subcutaneous injection of a RFID microchip (Planet ID GmbH) under brief Isoflurane inhalation anesthesia. Three cohorts comprising a total of 37 female mice were examined. For further details on behavioral tests and how they were conducted, see [9].

### 4.15. Statistical Analysis

Data from behavioral studies were analyzed using an ANOVA model with genotype (*MRPS5*^G315R/G315R^, *MRPS5*^WT/WT^) and age cohort (9, 14, 19 months) as between-subject factors. Within-subject factors were added as needed to explore the dependence of genotype effects on place, time, or stimulus. Interactions were further explored by pairwise *t*-tests or by splitting the ANOVA model, as appropriate. Variables with strongly skewed distributions or strong correlations between variances and group means were subjected to Box–Cox transformation before statistical analysis. The false discovery rate (FDR) control procedure of Hochberg was applied to groups of conceptually related variables within single tests to correct significance thresholds for multiple comparisons. Similarly, FDR correction was applied to *t*-tests during post hoc testing. Statistical analyses and graphs were produced using R version 3.2.3, complemented with the packages ggplot2, psych and moments. All tests were two-sided with significance set at *p* < 0.05.

### 4.16. Ethics Statement

Th animal experiments were approved by the Veterinary Office of the Canton of Zurich (licenses 29/2012 and 44/2015).

## Figures and Tables

**Figure 1 ijms-23-04384-f001:**
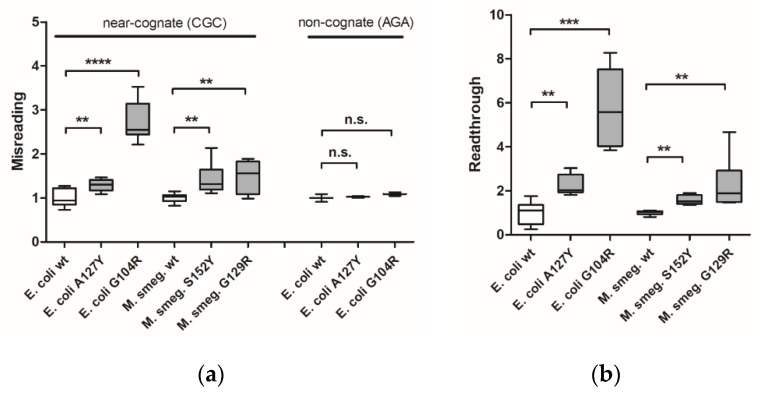
Measurement of translational accuracy in bacterial *ram* mutants. Misreading (**a**) and read-through (**b**) were measured by dual luciferase gain-of-function assays using *E. coli* wild-type ribosomes (*n* = 9 clones), *E. coli* mutant A127Y ribosomes (*n* = 9 clones), *E. coli* mutant G104R ribosomes (*n* = 9 clones), *M. smegmatis* merodiploid wild-type ribosomes (*n* = 9 clones), *M. smegmatis* merodiploid mutant S152Y ribosomes (*n* = 9 clones), and *M. smegmatis* merodiploid mutant G129R ribosomes (*n* = 9 clones). Results were derived by calculating mutant hFluc/hRluc activity related to wild-type hFluc/hRluc activity, wild-type samples were set as 1. Whiskers: min-max; ** *p* < 0.01, *** *p* < 0.001, **** *p* < 0.0001, n.s.—not significant.

**Figure 2 ijms-23-04384-f002:**
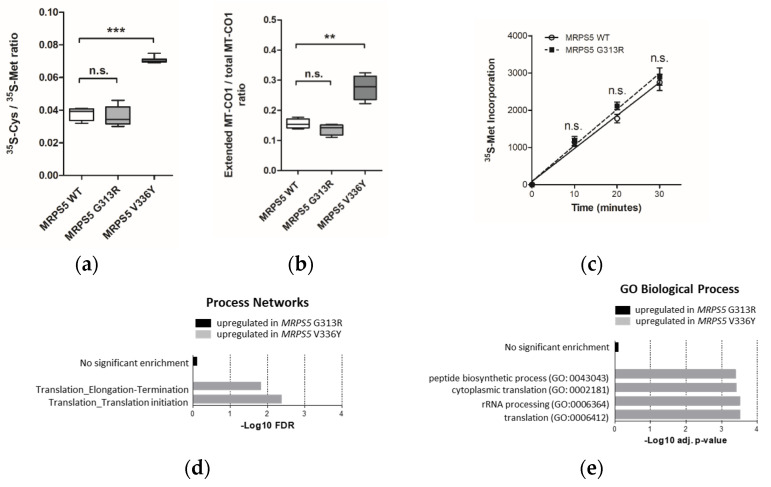
Analysis of HEK293 cells stably transfected with mutant *MRPS5* G313R or wild-type *MRPS5*. (**a**) Ratio of ^35^SCys and ^35^S-Met incorporation in MT-CO1 protein upon *in organello* translation. Cells transfected with *MRPS5* V336Y are used as a positive control (*n* = 8 clones). (**b**) Quantification of read-through of MT-CO1 protein in HEK293 cells by the ratio (poly-Lys stretch)-extended MT-CO1/total MT-CO1. Cells transfected with *MRPS5* V336Y are used as a positive control (*MRPS5* WT = 5 clones, *MRPS5* G313R = 7 clones, *MRPS5* V336Y = 7 clones). (**c**) In vivo mitochondrial translation as per ^35^S-Met incorporation (*n* = 4 clones). ** *p* < 0.01, *** *p* < 0.001, n.s.—not significant. (**d**,**e**) Transcriptome profiling analysis. (**d**) GO Biological Process term enrichment for significantly upregulated (adj. *p* < 0.05) genes in *MRPS5* mutants. Selected terms related to translation process are shown. (**e**) MetaCore Process Networks enrichment for significantly (FDR < 0.05) upregulated genes in *MRPS5* mutants. Selected terms related to translation process are shown.

**Figure 3 ijms-23-04384-f003:**
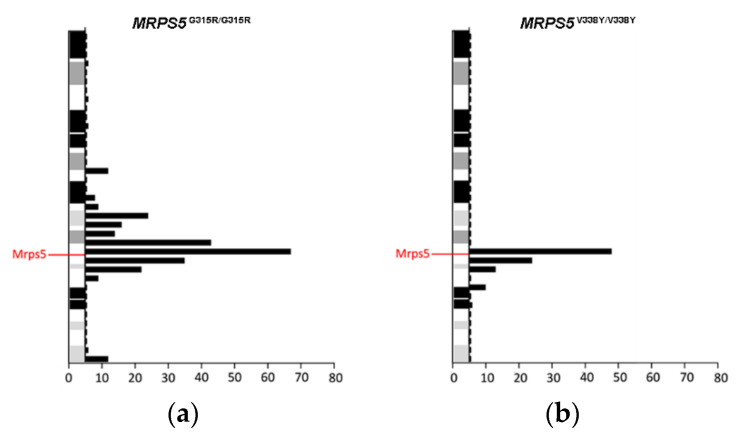
Distribution of Sv129-specific SNPs mapped within 5 Mb physical intervals on chromosome 2. (**a**) *MRPS5*^G315R/G315R^ mice. (**b**) *MRPS5*^V338Y/V338Y^ mice. SNPs representing the Sv129 genetic background were identified by comparison of total genomic cDNA sequences from corresponding mutant mice and wild-type C57BL/6 mice.

**Figure 4 ijms-23-04384-f004:**
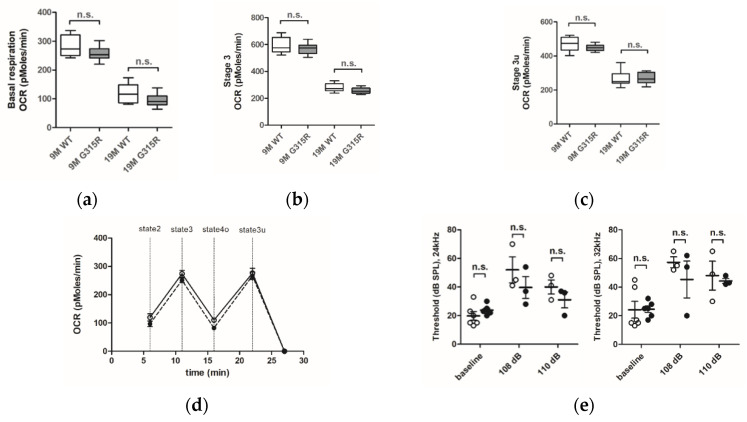
Effect of the mutation *MRPS5* G315R on oxygen consumption in the brain and noise-induced hearing damage. (**a**–**d**) Oxygen consumption rate (OCR) in the mouse cortex. (**a**) Baseline OCR in 9-month-old and 19-month-old mice. (**b**) State 3 OCR (after addition of ADP) in 9-month -old and 19-month-old mice. (**c**) state 3u (uncoupled) OCR (after addition of FCCP) in 9-month-old and 19-month-old mice. Whiskers: min-max, n.s., not significant. (**d**) Complete OCR profile in cortex mitochondria from 19-month-old mutant mice (black circle) compared with OCR from wild-type mice (white circle). The sequential injection of mitochondrial substrates and inhibitors is indicated by dotted lines (see details in the Methods section). Values corresponding to the different respiratory states are represented as mean ± SEM (*n* = 2 replicates of five animals per group). Two-way ANOVA revealed a significant effect of age (*p* < 0.0001), but not genotype (n.s., not significant). (**e**) Noise exposure and ABR measurements. Exposure of mice to sound levels of either 108 (left panel) or 110 dB (right panel) resulted in elevations of ABR thresholds both in *MRPS5*^WT/WT^ (open circles) and *MRPS5*^G315R/G315R^ (black circles) mice. Two-way ANOVA for combined 108 and 110 dB data (*n* = 6 for baseline littermates; *n* = 5 for baseline mutants; *n* = 3 for all noise exposures; ±SEM; n.s., not significant).

**Figure 5 ijms-23-04384-f005:**
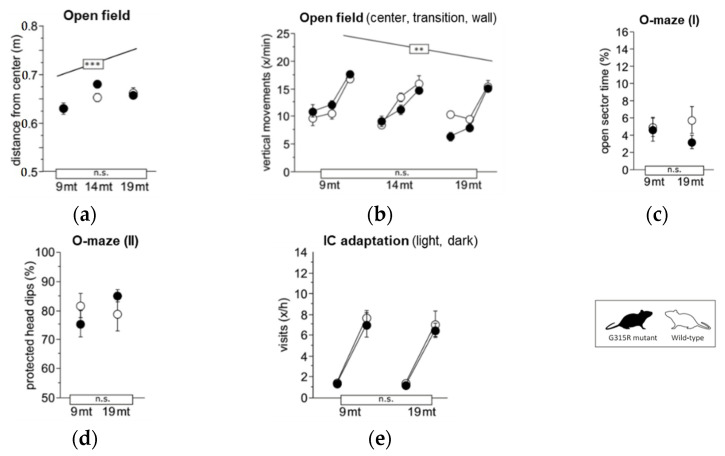
Exploration and anxiety. (**a**) Large open field, average distance from the center (ANOVA: genotype F1,34 = 1.088 ns, age F2,34 = 10.17 *p* = 0.0003, age × genotype F2,34 = 1.705 ns). (**b**) Large open field, estimated vertical movements per min spent in the center field, transition and wall zone (ANOVA: genotype F1,34 = 1.224 ns, age F2,34 = 6.152 *p* = 0.0052, zone F2,68 = 197.1 *p* < 0.0001, age × genotype F2,34 = 3.025 *p =* 0.0618, zone × genotype F2,68 = 0.320 ns, zone × age F4,68 = 4.889 *p* = 0.0016. (**c**) Elevated O-maze, time spent with all four paws on open sectors (ANOVA: genotype F1,35 = 1.425 ns, age F2,35 = 0.172 ns, age × genotype F2,35 = 0.206 ns). (**d**) Elevated O-maze, % protected head dips (genotype F1,34 = 1.611 ns, age F2,34 = 1.130 ns, age × genotype F2,34 = 0.780 ns). (**e**) Corner visits during free adaptation in the IntelliCage, counts per hour for the light and dark phase averaged over 6 days (ANOVA: genotype F1,36 = 0.854 ns, age F2,36 = 1.291 ns, phase F1,36 = 804.0 *p* < 0.0001, age × genotype F2,36 = 0.040 ns, phase × genotype F1,36 = 0.152 ns, phase × age F2,36 = 7.874 *p* = 0.0015). Graphs show mean ± SE. Boxes on the *x*-axis indicate significance of genotype, boxes above data points indicate significance of age, *** *p* < 0.001, ** *p* < 0.01, n.s. *p* ≥ 0.05. Mutants are depicted with black circles; wild-types with white circles. 34 mice, *n* = 17 per genotype, *n* = 9–13 per age cohort.

**Figure 6 ijms-23-04384-f006:**
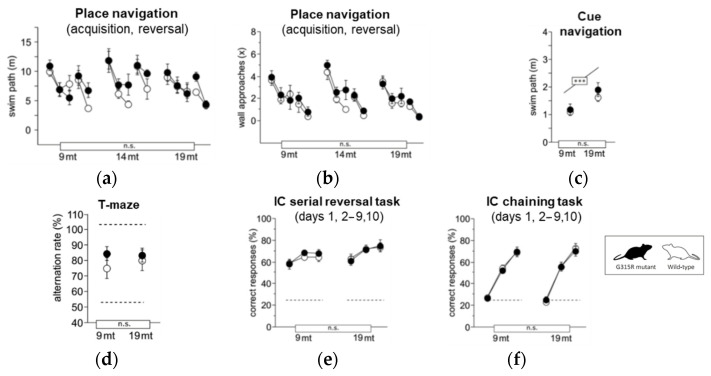
Learning and memory. (**a**) Water-maze place navigation, swim path during acquisition and reversal (ANOVA: genotype F1,33 = 1.832 ns, age F2,33 = 3.695 *p* = 0.0357, time F4,132 = 18.87 *p* < 0.0001, age × genotype F2,33 = 0.599 ns, time × genotype F4,132 = 1.496 ns, time × age F8,132 = 1.765 *p* = 0.0895). (**b**) Water-maze place navigation, number of wall approaches during acquisition and reversal training (ANOVA: genotype F1,33 = 1.310 ns, age F2,33 = 1.237 ns, time F4,132 = 55.43 *p* < 0.0001, age × genotype F2,33 = 0.282 ns, time × genotype F4,132 = 0.650 ns, time × age F8,132 = 0.426 ns). (**c**) Water-maze cue navigation, swim path averaged over two training days (ANOVA: genotype F1,33 = 1.034 ns, age F2,33 = 9.994 *p* = 0.0004, age × genotype F6,99 = 0.823 ns). (**d**) T-maze, % alternation averaged across six trial pairs (max. 100%, chance 50%) (ANOVA: genotype F1,35 = 0.322 ns, age F2,35 = 0.537 ns, age × genotype F2,35 = 0.874 ns). (**e**) IntelliCage serial corner reversal, % correct responses during days 1, 2–9 and 10 of training (ANOVA: genotype F1,32 = 0.460 ns, age F2,32 = 1.774 ns, time F2,64 = 20.02 *p* < 0.0001, age × genotype F2,32 = 0.152 ns, time × genotype F2,64 = 0.341 ns, time × age F4,64 = 0.330 ns). (**f**) IntelliCage chaining task, % correct responses during days 1, 2–9 and 10 of training (ANOVA: genotype F1,31 = 0.771 ns, age F2,31 = 2.290 ns, time F2,62 = 431.2 *p* < 0.0001, age × genotype F2,31 = 1.603 ns, time × genotype F2,62 = 0.406 ns, time × age F4,62 = 2.681 *p* = 0.0397). Graphs show mean and SE. Boxes on the *x*-axis indicate significance of genotype, boxes above data points indicate significance of age, *** *p* < 0.001, n.s. *p* ≥ 0.05. Mutants are depicted with black circles, wild-types with white circles. 34 mice, *n* = 17 per genotype, *n* = 9–13 per age cohort.

**Figure 7 ijms-23-04384-f007:**
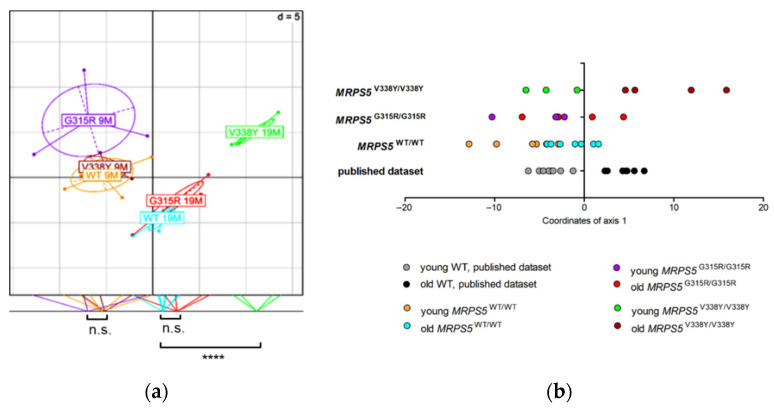
Muscle metabolome analysis of mice of 9 months and 19 months of age. (**a**) Between Group Analysis (BGA) scatter plot for muscle metabolite profiles (*n* = 5 for 9-month-old *MRPS5* WT; *n* = 3 for 9-month-old *MRPS5* G315R; *n* = 3 for 9-month-old *MRPS5* V338Y; *n* = 7 for 19-month-old *MRPS5* WT; *n* = 5 for 19-month-old *MRPS5* G315R; *n* = 4 for 19-month-old *MRPS5* V338Y). Coordinates of the animals as well as group centers along axis 1 are projected on the bottom line of the plot frame, and group differences are indicated (Welch’s *t*-test; **** *p* < 0.0001; n.s., not significant). (**b**) Analysis of common aging pattern between published WT study (Δold-young = 20 months), published V338Y mutant study (Δold-young = 10 months) and the present G315R mutant study (Δold-young = 10 months). The dataset consisted of 88 common metabolites between the datasets. The BGA of the published mouse dataset was calculated (training dataset). Mice of the V338Y and G315R datasets were used as test datasets, superimposed in the BGA and coordinates of axis 1 were plotted.

## Data Availability

Transcriptome data are available in the Gene Expression Omnibus (GEO), accession number GSE195772, token cfwfiksadvgpbgb. The complete original metabolome dataset is included in the Appendix A.

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
