# Peer review of "Phenotype of Mrps5-Associated Phylogenetic Polymorphisms Is Intimately Linked to Mitoribosomal Misreading"

_ijms, 2022, doi:10.3390/ijms23084384_

Round 1

Reviewer 1 Report

The manuscript describes a follow-on study to recently published work from the authors concerning increased mitochondrial translational error resulting from a mutation affecting mitochondrial ribosomal protein uS5m (V388Y) that they found associated with a variety of negative physiological effects including impaired mitochondrial function in the brain, reduced stress tolerance, anxiety effects and accelerated metabolic aging. The authors wanted to eliminate the idea that a marker linked to the gene encoding uS5m were the true cause of the effect. They chose to test whether another uS5m variant (G315R) that they predicted would not affect accuracy but derived from the same genetic background might share the physiological affects without altering error frequency.

The short answer is that G315R does not alter error frequency, measured both by a misreading and a translational readthrough assay in organello, and also fails to cause the V388Y physiological effects. They were careful to show that the two strains shared a similar region surrounding the primary mutation derived from the parental strain. They also show that G315R lacks any significant effects on well defined mitochondrial functions. The result seems extremely clear, if perhaps not surprising but it is laudable that the authors sought to eliminate so categorically a null hypothesis that the accuracy effects are not the cause of the physiological effects.

The study was well designed and performed and the description of the motivations and observations are extremely clear.

My only concern is that the use of several numbering strategies throughout the manuscript might be confusing. The numbering conventions for uS5 from bacterial, yeast, mouse and human proteins differ, in some cases significantly since the mammalian proteins are much longer. One possibility is to provide an alignment of the several proteins, as the authors do in Figure S1 but to chose to use one of the numbering conventions throughout, perhaps the human one since that is the true focus of the authors interest.

Author Response

Response: We thank the respective reviewer for the positive feedback.

To address the reviewers’ concern about using different numbering for the described mutations in ribosomal proteins i) we have added the complete alignment of ribosomal protein uS5 (mitochondrial, cytosolic and bacterial) as a Supplementary Figure S1a; a selected part of this alignment (former Supplementary Figure S1) becomes Supplementary Figure S1b; ii) we have changed throughout the whole manuscript the mouse numbering V338Y and G315R to the human V336Y and G313R where it was possible (i.e. we have kept mouse numbering V338Y in all cases where the genotype of the specific mouse line is mentioned since this mouse line was described as C57BL6 MRPS5V338Y/V338Y (Akbergenov et al., 2018) and changes in numbering might be confusing).

Enclosed please find the new version of Supplementary Figure 1a-b.

Reviewer 2 Report

The present article is devoted to the study of mitochondrial translation, an extremely important but little-studied process. The authors of the work have done a lot of experimental work, obtained many interesting results using various methods and approaches. Despite the fact that no significant effects on mitochondrial translation and mitochondrial functionality were observed as a result of the introduction of the G315R mutation in the Mrps5 protein, the data obtained are very important for understanding the molecular mechanisms of protein biosynthesis in mitochondria.
The text of the article is written in excellent scientific language, the methods used are clearly described, the data obtained are presented in full, the conclusions fully follow from the results obtained. The only remark should be the incorrect design of figures 2, 4, 5 and 6, where the panel captions have moved out.
I believe that the presented article can be published with minor revisions, since it is of significant scientific and methodological interest.

Author Response

Response: We thank the respective reviewer for the positive feedback.

The positions of panel captions of the Figures 2, 4, 5 and 6 have been corrected.

Reviewer 3 Report

Title: Phenotype of Mrps5-associated phylogenetic polymorphisms is intimately linked to mitoribosomal misreading

The authors have published mutant MRPS5 related 2 papers as following

  1. Mutant MRPS5 affects mitoribosomal accuracy and confers stress-related behavioral alterations
  2. Mitochondrial Mistranslation in Brain Provokes a Metabolic Response Which Mitigates the Age-Associated Decline in Mitochondrial Gene Expression

The authors described two results using bacteria

  1. Results

2.1. Modelling of ram mutations in bacterial RpsE, eukaryotic cytosolic Rps2 and mitochondrial Mrps5 ribosomal proteins

2.2. Mutations G104R and A127Y in RpsE confer misreading in bacterial ribosomes

The authors should explain how the results of bacteria experiments influence the experiments using mammalian cells.

Title of this article “Phenotype of Mrps5-associated phylogenetic polymorphisms is intimately linked to mitoribosomal misreading” exaggerates the contents of this article. The reviewer recommend to change to more specific title.

Figure 3: The authors should explain how these results influence the overall mouse results using mutant MRPS5.

Figure 4: The authors show no significant different results.

Figures 5 & 6: The authors show no significant different results.

Figure 7: The authors need to compare with the results published for brain metabolome analysis in 2021.

Author Response

Response: We thank the respective reviewer for the feedback.

Below is our point-by-point response to the reviewers’ comments (marked in red).

Title: Phenotype of Mrps5-associated phylogenetic polymorphisms is intimately linked to mitoribosomal misreading

The authors have published mutant MRPS5 related 2 papers as following

  1. Mutant MRPS5 affects mitoribosomal accuracy and confers stress-related behavioral alterations
  2. Mitochondrial Mistranslation in Brain Provokes a Metabolic Response Which Mitigates the Age-Associated Decline in Mitochondrial Gene Expression

Note: there is a third related publication, namely “Shcherbakov D, Duscha S, Juskeviciene R, Restelli L, Frank S, Laczko E, et al. Mitochondrial misreading in skeletal muscle accelerates metabolic aging and confers lipid accumulation and increased inflammation. RNA. 2020, 27(3), 265-72” cited in the manuscript.

The authors described two results using bacteria

  1. Results

2.1. Modelling of ram mutations in bacterial RpsE, eukaryotic cytosolic Rps2 and mitochondrial Mrps5 ribosomal proteins

2.2. Mutations G104R and A127Y in RpsE confer misreading in bacterial ribosomes

The authors should explain how the results of bacteria experiments influence the experiments using mammalian cells.

The manuscript is primarily focused on mutation G313R in the mitochondrial protein Mrps5. This mutation has been identified as homologous to the described bacterial RpsE G104R ram mutation, however in contrast to RpsE G104R the mutation Mrps5 G313R does not confer misreading. We have demonstrated both effects with a bacterial in vitro system and a mitochondrial in organello system and shown that the discrepancy can be explained on the basis of structural modelling. To support this approach, we have shown that the mistranslation caused by the described ram mutations RpsE A127Y (E. coli) and Mrps5 V336Y (human mitochondria) is in agreement with the results of structural modelling.

Title of this article “Phenotype of Mrps5-associated phylogenetic polymorphisms is intimately linked to mitoribosomal misreading” exaggerates the contents of this article. The reviewer recommend to change to more specific title.

With all due respect, we disagree; in our opinion the title properly reflects the main result of the manuscript.

Figure 3: The authors should explain how these results influence the overall mouse results using mutant MRPS5.

The requested explanation is actually provided in the Discussion, lines 420-434, as follows:

“Our data obtained with transfected HEK293 human cell lines provide strong evidence that Mrps5 G313R in contrast to V336Y is not a mitochondrial ram mutation. To further challenge the link between V336Y-mediated mitochondrial misreading and the phenotype observed in transgenic MRPS5V338Y/V338Y mice, we generated homozygous knock-in mice MRPS5G315R/G315R. In particular, we wished to exclude i) that mutations in MRPS5 affect mitochondrial function independently of misreading, and ii) that it is not the V338Y mutation, but the surrounding Sv129 genomic locus which is responsible for the MRPS5V338Y/V338Y mice phenotype. As per classical transgenic mice technology, the MRPS5 mutations were introduced into embryonic stem (ES) cells of Sv129 genetic background and then transferred to a C57BL/6 background by successive backcrossing. Inherently, the genomic region flanking the targeted gene in mutant (but not WT littermate control) mice was carried from the Sv129 background during backcrossing and might affect the phenotypic outcome [30]. RNA-seq data revealed that the size of the Sv129-derived donor region in MRPS5V338Y/V338Y and MRPS5G315R/G315R mice was similar, with approximately 25 Mb and 50 Mb, respectively.”

Figure 4: The authors show no significant different results.

Correct. See below.

Figures 5 & 6: The authors show no significant different results.

Correct. The absence of significant differences in the experiments illustrated in Figures 4-6 is summarized in the Discussion, lines 435-447, as follows:

MRPS5G315R/G315R mice were subjected to a set of behavioral tests similar to that used previously for characterization of the MRPS5V338Y/V338Y mice [9]. Our investigations in G315R mice did not show any of the phenotypic changes characteristic for the MRPS5V338Y/V338Y mutant mice, such as stress intolerance or anxiety-related behavioral alterations, nor any other phenotypic trait different from WT controls. In addition, neither did we observe increased susceptibility to noise-induced hearing loss. Importantly, we also did not find any effect of the Mrps5 G315R mutation on mitochondrial respiration, ATP levels or generation of ROS. Furthermore, metabolic profiles of skeletal muscle tissue from MRPS5G315R/G315R mice were not significantly different from those of wild-type mice. Besides testifying to the specific effects of the V338Y mutation, the absence of phenotypic alterations in the G315R mutants effectively rules out the possibility that the Sv129 genomic locus encompassing the Mrps5 mutation confers the phenotypic alterations observed in the V338Y mutants.”

This lack of phenotypic effect for the mutation Mrps5 G313R is a main focus of the manuscript as reflected in the conclusion (lines 448-454):

“In conclusion, we here provide evidence that mutation Mrps5 G313R, the mitochondrial homologue of the bacterial RspE G104R ram mutation, does not affect the accuracy of mitochondrial translation and does not show any of the phenotypic changes observed in MRPS5V338Y/V338Y transgenic mice. Our results strengthen the pathomechanistic link between error-prone mitochondrial translation, impaired mitochondrial respiration, enhanced susceptibility to noise-induced hearing damage, anxiety-related behavioral alterations, and accelerated age-related metabolic changes in muscle of MRPS5V338Y/V338Y mice.”

Figure 7: The authors need to compare with the results published for brain metabolome analysis in 2021.

Comparing the metabolomic profile of such different organs as skeletal muscle and brain makes very little sense. Unfortunately, we have no metabolomic data for the brains of mutant Mrps5 G315R mice. However, the metabolomic data for the skeletal muscle from those mice are available and in the manuscript we compared metabolomic profiles of the skeletal muscles from Mrps5 G315R (this work) and Mrps5 V338Y (Shcherbakov D, Duscha S, Juskeviciene R, Restelli L, Frank S, Laczko E, et al. Mitochondrial misreading in skeletal muscle accelerates metabolic aging and confers lipid accumulation and increased inflammation. RNA. 2020, 27(3), 265-72). The results are described in lines 377-389 and illustrated by Figure 7.

Round 2

Reviewer 3 Report

  • This mutation has been identified as homologous to the described bacterial RpsE G104R ram mutation, however in contrast to RpsE G104R the mutation Mrps5 G313R does not confer misreading. We have demonstrated both effects with a bacterial in vitro system and a mitochondrial in organello system and shown that the discrepancy can be explained on the basis of structural modelling. To support this approach, we have shown that the mistranslation caused by the described ram mutations RpsE A127Y ( coli) and Mrps5 V336Y (human mitochondria) is in agreement with the results of structural modelling.

→ I understood. Why don’t you add these sentences in Discussion.

Author Response

We apologize for not having made clear how the results of bacteria experiments influence the experiments using mammalian cells and appreciate the reviewer's comment guiding our attention to this point.

As per the reviewer's suggestion the additional clarification was introduced into Discussion as following (lines 408-415):

“In other words, although the mutation Mrps5 G313R has been identified as homologous to the described bacterial RpsE G104R ram mutation, in contrast to RpsE G104R the mutation Mrps5 G313R does not confer misreading. Here we have demonstrated both effects with a bacterial in vitro system and a mitochondrial in organello system and shown that the discrepancy can be explained on the basis of structural modelling. To support this approach, we have shown that the mistranslation caused by the described ram mutations RpsE A127Y (E. coli) and Mrps5 V336Y (human mitochondria) is in agreement with the results of structural modelling.”